# OBJECT-CENTRIC PRETRAINING VIA TARGET ENCODER BOOTSTRAPPING

**Nikola Đukić**     **Tim Lebailly**     **Tinne Tuytelaars**
KU Leuven
{nikola.djukic, tim.lebailly, tinne.tuytelaars}@kuleuven.be

## ABSTRACT

Object-centric representation learning has recently been successfully applied to real-world datasets. This success can be attributed to pretrained non-object-centric foundation models, whose features serve as reconstruction targets for slot attention. However, targets must remain frozen throughout the training, which sets an upper bound on the performance object-centric models can attain. Attempts to update the target encoder by bootstrapping result in large performance drops, which can be attributed to its lack of object-centric inductive biases, causing the object-centric model's encoder to drift away from representations useful as reconstruction targets. To address these limitations, we propose **O**bject-**CE**ntric Pretraining by Target Encoder **BO**otstrapping, a self-distillation setup for training object-centric models from scratch, on real-world data, for the first time ever. In OCEBO, the target encoder is updated as an exponential moving average of the object-centric model, thus explicitly being enriched with object-centric inductive biases introduced by slot attention while removing the upper bound on performance present in other models. We mitigate the slot collapse caused by random initialization of the target encoder by introducing a novel cross-view patch filtering approach that limits the supervision to sufficiently informative patches. When pretrained on 241k images from COCO, OCEBO achieves unsupervised object discovery performance comparable to that of object-centric models with frozen non-object-centric target encoders pretrained on hundreds of millions of images. The code and pretrained models are publicly available at https://github.com/djukicn/ocebo.

## 1 INTRODUCTION

In recent years, large-scale foundation models have become ubiquitous across many domains of deep learning. This is mainly thanks to the development of self-supervised learning (SSL) techniques that enable pretraining on massive amounts of unlabeled data. In computer vision, several families of SSL techniques have emerged, focusing on (i) imposing cross-view invariance at the global (Grill et al., 2020; Caron et al., 2021), patch (Zhou et al., 2022; Stegmüller et al., 2023) or cluster (Wen et al., 2022; Lebailly et al., 2024) level, or (ii) reconstruction in the pixel (He et al., 2021) or latent space (Zhou et al., 2022; Assran et al., 2023). On the other hand, research in cognitive psychology (Benjamin Peters, 2021) suggests that human visual perception reasons about visual inputs by decomposing scenes into sets of objects with corresponding representations. In the past years, efforts have been made to mimic this behavior (Locatello et al., 2020; Seitzer et al., 2022; Kakogeorgiou et al., 2024; Didolkar et al., 2024) and move past global and patch representations towards object-centric representations. Object-centric models are not only more biologically plausible (Benjamin Peters, 2021), but can provably achieve compositional generalization (Wiedemer et al., 2024) and have already proven useful on several downstream tasks, such as robotic control (Zadaianchuk et al., 2021; Haramati et al., 2024), compositional generation (Singh et al., 2022; Wu et al., 2023; Jiang et al., 2023; Sajjadi et al., 2022), and visual question answering (Xu et al., 2024). While incredibly promising, object-centric models have not yet been successfully pretrained on a large-scale real-world dataset.

A slot attention-based object-centric model consists of an image encoder, slot attention encoder and slot decoder. Initially, these were trained end-to-end with a reconstruction objective in the pixel-

space (Locatello et al., 2020), resulting in good scene decomposition on synthetic datasets but slot collapse [1] on real-world data. Seitzer et al. (2022) introduce a reconstruction objective in the latent space of a pretrained encoder (e.g., DINO (Caron et al., 2021), DINOv2 (Oquab et al., 2023), etc.), showing that the presence of informative reconstruction targets is crucial for avoiding slot collapse. As this encoder provides reconstruction targets, we refer to it as a *target encoder*. Moreover, Seitzer et al. (2022) initialize the encoder of their object-centric model with the same pretrained encoder and keep it frozen alongside the target encoder, effectively training only the slot attention encoder and decoder.

While the framework with the frozen target encoder achieves unprecedented performance on real-world datasets (Seitzer et al., 2022), it has a significant limitation. Namely, the target encoder cannot be updated, meaning that there exists an upper bound on what the object-centric model can learn from it. In fact, despite the target encoder being pretrained on millions or even hundreds of millions of images (Caron et al., 2021; Oquab et al., 2023), the object-centric model can efficiently consume all its knowledge and reach the upper limit. This is demonstrated in an experiment by Didolkar et al. (2024), where the performance plateaus with ∼16k images from MS COCO (Lin et al., 2015) and does not further increase, even with an order of magnitude more data. This clashes with the scalability trends observed in other (self-supervised) representation learning families. A question arises — *how to update the target encoder, thus removing the performance upper limit and enabling large-scale pretraining of object-centric models?*

Kakogeorgiou et al. (2024) show for the first time that unfreezing a few final layers of the object-centric model's encoder is possible and further improves the performance. Didolkar et al. (2024) further extend this finding by fine-tuning the full encoder. A straightforward approach would be updating the target encoder as an exponential moving average (EMA) of the object-centric model's encoder. However, this has been shown to result in huge performance drops (Didolkar et al., 2024). We hypothesize that the reason for this is the *lack of object-centric inductive biases* in the target encoder. Most often trained for cross-view consistency, SSL models organize their representations in terms of semantics, rather than object instances. For example, semantically similar parts of different objects will be close in the representation space and farther from less similar parts of the same object. When the object-centric model tries to reconstruct such features, its inductive bias focuses on assigning these semantically similar regions to different objects (i.e., instances), thus updating its encoder to focus slightly less on semantics and more on positions in the image. EMA update of the target encoder then results in informative knowledge slowly leaking at the cost of more positional information, which ultimately degrades the performance.

In this work, we argue that a way to overcome the current limitations of object-centric learning and unlock its full potential is by large-scale pretraining from scratch. We pose object-centric learning as a self-distillation bootstrapping problem (Caron et al., 2021), in which the object-centric model is distilled from the target encoder that is in turn updated as an EMA of the object-centric model's encoder. We propose the *object-centric self-distillation loss*, a patch-level loss that acts as a reconstruction objective. We apply it in a cross-view consistency fashion to enrich slot attention with augmentation invariance. To prevent slot collapse stemming from the lack of informative reconstruction targets due to random initialization of the target encoder, we propose a *cross-view patch filtering* procedure that uses cross-view correspondences to determine which patches to reconstruct at a particular training stage. The resulting model, named OCEBO allows pretraining object-centric models on real-world datasets for the first time. The EMA updates not only remove the upper bound from object-centric models but explicitly introduce object-centric inductive biases into the target encoder. By training from scratch, we allow the model to learn informative instance-based features rather than trying to reorganize semantic features from a pretrained non-object-centric encoder. When pretrained on ∼118k or ∼241k images from MS COCO (Lin et al., 2015), not only does OCEBO avoid slot collapse, but also achieves performance comparable to that of object-centric models with a DINOv2 (Oquab et al., 2023) target encoder pretrained on 142M images, proving the object-centric inductive biases highly beneficial for the target encoder. Moreover, OCEBO demonstrates scalability well beyond a few thousand training images, unlike other object-centric models. Altogether, we believe this work paves a way towards *large-scale pretraining of object-centric foundation models*.

---

[1]Slot collapse refers to a scenario where slots learn to attach to positionally coherent blocks rather than actual objects.

## 2 RELATED WORK

**Self-distillation.** Cross-view consistency has emerged as a predominant self-supervised learning (SSL) task in computer vision. In this framework, two views of an input image, obtained through data augmentations, are used to enforce cross-view representation consistency while avoiding representation collapse. While some works avoid collapse via the use of negative samples (Chen et al., 2020; He et al., 2020), self-distillation approaches rely on positive samples only, in combination with mechanisms such as branch asymmetry (Grill et al., 2020), stop-gradients (Chen & He, 2021; Caron et al., 2021), momentum encoders (Grill et al., 2020; Caron et al., 2021), etc. Moreover, self-distillation methods can enforce consistency at different levels of granularity. In global or image-level self-supervision (Grill et al., 2020; Caron et al., 2021), consistency is enforced between global representations that capture the whole input image. In patch-level or local self-supervision (Zhou et al., 2022; Stegmüller et al., 2023), consistency is enforced between pairs of corresponding patches in both views. Finally, consistency has been enforced between semantically coherent groups of patches (Wen et al., 2022; Lebailly et al., 2024), i.e., clusters (also referred to as cluster-level or object-level self-supervision).

**Object-centric representation learning.** Image-, patch- and cluster-level self-distillation methods have achieved impressive performance across computer vision. In parallel, attempts at object-centric representation learning have been made. An object-centric model decomposes a scene into a set of objects with corresponding representations, which is in line with theories about underlying mechanisms of human visual perception (Benjamin Peters, 2021). Unlike object-level self-distillation methods that rely on soft clustering (Wen et al., 2022; Lebailly et al., 2024), which can often group semantically similar parts of different instances, object-centric models explicitly use inductive biases that enforce instance-level scene decomposition. Slot attention (Locatello et al., 2020) has emerged as a particularly successful inductive bias. It introduces an attention mechanism that promotes competition among slots for different parts of input. In the original work (Locatello et al., 2020), an autoencoding model consisting of an image encoder, slot encoder and slot decoder trained end-to-end has been proposed. It achieved impressive performance on well-structured synthetic datasets but resulted in slot collapse on real-world datasets. Seitzer et al. (2022) replaced the pixel space reconstruction objective with reconstruction of features produced by a frozen pretrained non-object-centric SSL model, enabling the training of object-centric models on real-world data. Subsequent works focused on different decoding strategies (Kakogeorgiou et al., 2024) or fine-tuning the encoder while keeping the target encoder frozen (Didolkar et al., 2024). Diffusion-based objectives Wu et al. (2023); Jiang et al. (2023) have been proposed as an alternative to reconstruction-based objective. However, these still rely on pretrained SSL models. To the best of our knowledge, direct pretraining of object-centric models on real-world datasets has not been achieved yet.

Note that SlotCon (Wen et al., 2022) also introduces the concept of slots. However, these serve as learnable queries whose assignment to patch representations of both views is used as a supervision signal. In this context, a larger number of slots corresponding to the number of object types in the training dataset is initialized, such that each slot learns to represent a particular object type. In scenes with multiple instances of a same object type, SlotCon would not produce one representation per instance, but rather a single slot attached to patches of all instances. This is fundamentally different from OCEBO and other object-centric works, where instance-level decomposition is achieved.

## 3 METHOD

An overview of OCEBO is provided in Figure 1. We provide the necessary background in Section 3.1, continuing with the training objective in Section 3.2, cross-view patch filtering approach in Section 3.3 and the optional mask sharpening stage in Section 3.4.

### 3.1 PRELIMINARIES

**Image views.** Let $x \in \mathbb{R}^{h \times w \times c}$ be an image from the training dataset. We obtain two views $x_1, x_2 \in \mathbb{R}^{h \times w}$ by applying two randomly sampled sets of data augmentations to the original image $x$. When sampling augmentations, we ensure that there exists some overlap between $x_1$ and $x_2$. We define the inverse augmentation function $\texttt{invaug} : \mathbb{R}^{h \times w \times c} \mapsto \mathbb{R}^{h \times w \times c}$, such that $\texttt{invaug}(x_1)$ and $\texttt{invaug}(x_2)$ cut the overlapping regions from views $x_1$, $x_2$ and interpolate them back to size

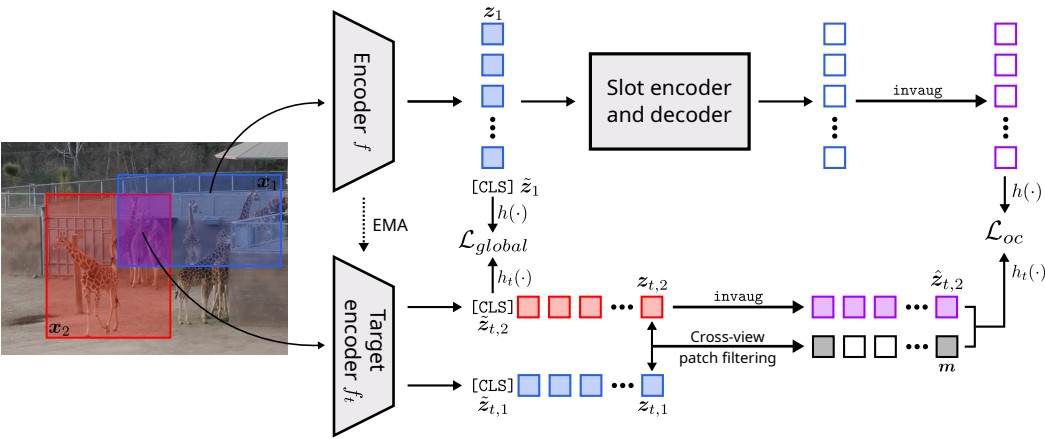

Figure 1: Overview of OCEBO: View $x_1$ is processed by the object-centric model's encoder (top branch), producing global and patch representations $\tilde{z}_1$ and $z_1$, respectively. Patch representations are sent through the slot attention encoder and decoder, where the latter outputs a reconstruction of the input patch representations $q_1$. Target encoder (bottom branch) processes both views $x_1$ and $x_2$ separately and produces their global and patch representations $\tilde{z}_{t,1}$, $z_{t,1}$, $\tilde{z}_{t,2}$ and $z_{t,2}$, respectively. Patch representations $z_{t,1}$ and $z_{t,2}$ are used by the cross-view patch filtering approach to infer informative target patches and produce the mask $m$. The inverse augmentation operation (invaug) is applied to the target features of $x_2$ and reconstructions of $x_1$ to make them correspond to the overlapping region (purple part of the image) before combining them with the mask $m$ and applying the object-centric loss $\mathcal{L}_{oc}$. Global loss $\mathcal{L}_{global}$ is applied to global representations $\tilde{z}_1$ and $\tilde{z}_{t,2}$.

$h \times w \times c$ (see Wen et al. (2022) for more details). We denote outputs of the invaug operation with a hat, e.g., invaug($\boldsymbol{x}_1$) = $\hat{\boldsymbol{x}}_1$.

**Object-centric model.** The encoder $f$ of our object-centric model is a Vision Transformer (ViT) (Dosovitskiy et al., 2021). For a view $\boldsymbol{x}_1$, $f$ outputs a global image representation (i.e., the [CLS] token) $\tilde{z}_1 \in \mathbb{R}^d$ and patch representations $z_1 \in \mathbb{R}^{N \times d}$, where $d$ is the embedding dimension, $p$ is the patch size, and $N = h/p \times w/p$ is the number of patches. Patch representations $z_1$ are then processed by a slot attention encoder, which maps them into $s$ $d_s$-dimensional slots. These are finally decoded by a slot decoder, resulting in a reconstruction $\boldsymbol{q}_1 \in \mathbb{R}^{N \times d}$ of the original input patch representations. Similarly, we define $\tilde{z}_2$, $z_2$ and $\boldsymbol{q}_2$ for the view $\boldsymbol{x}_2$. We also the define the inverse augmentation operation for representations. For instance invaug($\boldsymbol{q}_1$) cuts the features of $\boldsymbol{q}_1$ according to the corresponding overlapping region of $\boldsymbol{x}_1$ and $\boldsymbol{x}_2$ and interpolates them back to the original size of $\boldsymbol{q}_1$. Applying this operation to representations ensures they represent only the overlapping region and are aligned.

**Target encoder.** Target encoder $f_t$ has the identical architecture as the encoder $f$ of the object-centric model. Similarly to $f$, $f_t$ outputs global and patch representations $\tilde{z}_{t,1}$, $\tilde{z}_{t,2}$, $z_{t,1}$ and $z_{t,2}$. Initially, the parameters of $f_t$ are the same as the parameters of $f$. After each update of the object-centric model, the parameters of $f_t$ are modified as an exponential moving average (EMA) of the parameters of $f$.

## 3.2 OBJECT-CENTRIC TRAINING OBJECTIVE

Our training framework is formulated as a self-distillation bootstrapping problem (Caron et al., 2021), where the object-centric model can be viewed as the student and the target encoder as the teacher. To this end, we introduce projection heads $h$ and $h_t$, where $h_t$ is updated as an EMA of $h$. We define the object-centric self-distillation loss as

$$\mathcal{L}_{ocp} = \frac{1}{2}\left(\sum_{i=1}^{N} H(\boldsymbol{p}_{t,1}, \boldsymbol{p}_1) + \sum_{i=1}^{N} H(\boldsymbol{p}_{t,2}, \boldsymbol{p}_2)\right), \tag{1}$$

where $H(\boldsymbol{a}, \boldsymbol{b}) = \sum_j -\boldsymbol{a}_j \log \boldsymbol{b}_j$ and $\boldsymbol{p}_{t,1}, \boldsymbol{p}_{t,2}, \boldsymbol{p}_1, \boldsymbol{p}_2$ are sharpened (and centered) $L$-dimensional distributions obtained from outputs of the target encoder and the object-centric model subsequently mapped by projection heads, i.e.,

$$\boldsymbol{p}_{\{1,2\}} = \underset{L}{\text{softmax}}\left(h(\boldsymbol{q}_{\{1,2\}})/\tau\right)$$
$$\boldsymbol{p}_{t,\{1,2\}} = \underset{L}{\text{softmax}}\left(\left(h_t(\boldsymbol{z}_{t,\{1,2\}}) - \boldsymbol{c}_t\right)/\tau_t\right) \tag{2}$$

with temperature parameters $\tau$ and $\tau_t$ controlling the scaling and $\boldsymbol{c}_t$ controlling the centering.

Note that in most self-distillation scenarios, student and teacher have identical architectures. To avoid a trivial solution, cross-entropy is applied between features of two different views rather than within the same view as in equation 1. In our case, this is possible because the target encoder does not have the slot attention encoder and decoder. Nonetheless, by imposing $\mathcal{L}_{ocp}$ across views, we enforce augmentation invariance, which improves generalization of the object-centric model. We use the `invaug` operation to ensure that we apply the loss only in the intersection of views $\boldsymbol{x}_1$ and $\boldsymbol{x}_2$. Equations 1 and 2 then become

$$\mathcal{L}_{oc} = \frac{1}{2}\left(\sum_{i=1}^{N} H(\boldsymbol{p}_{t,1}, \boldsymbol{p}_2) + \sum_{i=1}^{N} H(\boldsymbol{p}_{t,2}, \boldsymbol{p}_1)\right), \tag{3}$$

and

$$\boldsymbol{p}_{\{1,2\}} = \underset{L}{\text{softmax}}\left(\text{invaug}\left(h(\boldsymbol{q}_{\{1,2\}})\right)/\tau\right)$$
$$\boldsymbol{p}_{t,\{1,2\}} = \underset{L}{\text{softmax}}\left(\left(\text{invaug}\left(h_t(\boldsymbol{z}_{t,\{1,2\}})\right) - \boldsymbol{c}_t\right)/\tau_t\right). \tag{4}$$

Similarly to other patch-level self-distillation works (Zhou et al., 2022; Stegmüller et al., 2023; Lebailly et al., 2024), we apply an additional cross-view global loss introduced by Caron et al. (2021) to ensure the training stability and improve the quality of representations. We define

$$\mathcal{L}_{global} = \frac{1}{2}\left(\sum_{i=1}^{N} H(\tilde{\boldsymbol{p}}_{t,1}, \tilde{\boldsymbol{p}}_2) + \sum_{i=1}^{N} H(\tilde{\boldsymbol{p}}_{t,2}, \tilde{\boldsymbol{p}}_1)\right), \tag{5}$$

where

$$\tilde{\boldsymbol{p}}_{\{1,2\}} = \underset{L}{\text{softmax}}\left(h(\tilde{\boldsymbol{z}}_{\{1,2\}})/\tau\right)$$
$$\tilde{\boldsymbol{p}}_{t,\{1,2\}} = \underset{L}{\text{softmax}}\left(\left(h_t(\tilde{\boldsymbol{z}}_{t,\{1,2\}}) - \tilde{\boldsymbol{c}}_t\right)/\tau_t\right). \tag{6}$$

The final training objective is $\mathcal{L} = \lambda_{oc}\mathcal{L}_{oc} + \lambda_{global}\mathcal{L}_{global}$.

## 3.3 CROSS-VIEW PATCH FILTERING

Previous works (Seitzer et al., 2022; Didolkar et al., 2024) have shown that avoiding slot collapse requires reconstruction targets of good quality. However, the features from our randomly initialized target encoder do not satisfy this. We propose the cross-view patch filtering condition as a proxy for the quality of target encoder features and apply the object-centric loss only to the patches that satisfy it. Intuitively, if we take a representation of a particular patch in one view and compute its distance to all patches of the other view, its nearest neighbor should be the exact same patch in the other view. If the target encoder's representations do not capture this, we do not use them in $\mathcal{L}_{oc}$. We allow some slack in the target encoder and consider $k$ nearest neighbors instead of just one. More formally, we define a binary mask $\boldsymbol{m} \in \{0, 1\}^N$, whose $i$-th entry is defined as

$$\boldsymbol{m}_i = \begin{cases} 1 & \text{if } i \in \underset{k}{\mathrm{nns}}(\hat{\boldsymbol{z}}_{t,1}, \hat{\boldsymbol{z}}_{t,2})_i \wedge i \in \underset{k}{\mathrm{nns}}(\hat{\boldsymbol{z}}_{t,2}, \hat{\boldsymbol{z}}_{t,1})_i \\ 0 & \text{otherwise,} \end{cases} \tag{7}$$

where $\mathrm{nns}_k(\hat{\boldsymbol{z}}_{t,1}, \hat{\boldsymbol{z}}_{t,2})_i$ denotes indices of $k$ nearest neighbors of the $i$-th patch of $\hat{\boldsymbol{z}}_{t,1}$ in $\hat{\boldsymbol{z}}_{t,2}$. Note that $\hat{\boldsymbol{z}}_{t,1} = \mathrm{invaug}(\boldsymbol{z}_{t,1})$ and $\hat{\boldsymbol{z}}_{t,2} = \mathrm{invaug}(\boldsymbol{z}_{t,2})$.

Incorporating the mask $\boldsymbol{m}$ into equation 3 yields

$$\mathcal{L}_{oc} = \frac{1}{2} \left( \sum_{i=1}^{N} \boldsymbol{m}_i H(\boldsymbol{p}_{t,1}, \boldsymbol{p}_2) + \sum_{i=1}^{N} \boldsymbol{m}_i H(\boldsymbol{p}_{t,2}, \boldsymbol{p}_1) \right). \tag{8}$$

In the initial epochs of training, the entries in $\boldsymbol{m}$ are mostly 0 and they keep increasing throughout the training. This allows the model to learn informative features and prevents it from collapsing (see Section 4.2).

### 3.4 MASK SHARPENING STAGE

Due to a constantly changing target encoder, we sometimes observe a lack of clear boundaries in the masks produced by OCEBO. We mitigate this by introducing an optional shorter mask sharpening stage, in which we keep training the object-centric model with a frozen target and an $\ell_2$ reconstruction loss instead of the self-distillation loss. This can be viewed as a procedure similar to fine-tuning of FT-DINOSAUR (Didolkar et al., 2024), but with a specialized target encoder enriched with object-centric inductive biases instead of the non-object-centric DINOv2 target encoder. After the sharpening step, masks exhibit clearer boundaries, resulting in improved unsupervised object-discovery performance (see Section 4.2 for details).

## 4 EXPERIMENTS

We start by outlining the implementation details (training datasets, evaluation protocols, model architecture and the training setup) of OCEBO in Section 4.1. In Section 4.2, we demonstrate that OCEBO can be pretrained from scratch on real-world data without slot collapse. We justify the design choices and demonstrate data scalability, further discussing the requirements for suitable pretraining datasets. Finally, we put the performance of OCEBO in context by comparing it to state-of-the-art object-centric approaches that rely on non-object-centric target encoders pretrained on orders of magnitude more data in Section 4.3. In addition, in Appendix A we demonstrate that OCEBO also produces high-quality patch representations, which is an important step towards versatile backbones capable of performing well in both object- and patch-level downstream tasks.

### 4.1 IMPLEMENTATION DETAILS

**Training data.** OCEBO is trained on MS COCO (Lin et al., 2015), the most common real-world dataset in the object-centric literature. We use the `train2017` COCO split with approximately 118k images. Additionally, we construct a larger dataset of ∼241k images named COCO+ by combining the `train2017` and `unlabeled2017` splits. We follow the standard data augmentation procedure with random crops resized to $224 \times 224$, horizontal flips, solarization, blurring and color jitter (Caron et al., 2021).

**Evaluation protocols.** Current trends in object-centric learning define in-distribution unsupervised object discovery as the go-to evaluation task for object-centric models (Seitzer et al., 2022; Kakogeorgiou et al., 2024). This means that a model is trained on a training split of a dataset and evaluated on the test split of the very same dataset. We believe this to be suboptimal as it does not agree with other tasks in computer vision. Moreover, OCEBO is trained from scratch and pretraining it on a small synthetic dataset is not a possibility. We believe the recently proposed zero-shot unsupervised object discovery (Didolkar et al., 2024) to be a more natural and fair way of evaluating. In this context, the models are trained on COCO and evaluated on 7 natural and synthetic datasets – MOVi-C and MOVi-E (Greff et al., 2022), ScanNet (Dai et al., 2017), YCB (Calli et al., 2015), ClevrTex (Karazija et al., 2021), Pascal VOC (Everingham et al., 2010) and EntitySeg (Qi et al.,

2023). For brevity, we report results on two natural and two synthetic datasets – MOVi-C, MOVi-E, Pascal VOC and EntitySeg. We find the conclusions on the remaining three synthetic datasets identical. We use validation splits of each dataset and 11, 24, 7 and 7 slots for MOVi-C, MOVi-E, Pascal VOC and EntitySeg, respectively. We report the foreground-Adjusted Rand Index (FG-ARI) and mean Best Overlap (mBO) as evaluation metrics. ARI treats ground truth and predicted masks as clustering assignments and measures their similarity. FG-ARI does the same, albeit on foreground pixels only. The mBO metric assigns to each ground truth mask a predicted mask with the highest overlap and reports the mean Intersection over Union (IoU) between the assignments.

**Architecture.** Both $f$ and $f_t$ follow the ViT-S/16 (Dosovitskiy et al., 2021) architecture. The slot attention encoder operates on $s = 7$ slots during training and consists of 3 layers with $d_s = 256$. Each layer consists of slot attention (Locatello et al., 2020) with a single attention head, a Gated Recurrent Unit and a 2-layer MLP with hidden dimension of 1024. Slot decoder is a 3-layer MLP with the hidden dimension of 2048. The projection heads are identical to those of DINO (Caron et al., 2021), with the exception of setting $L = 8192$ instead of the original 65536. Compared to the DINO head, ours projects every patch rather than just the global representations and we find that the gain in performance does not justify the computational cost.

**Optimization.** We train the model for 300 epochs with an additional mask sharpening stage of 100 epochs. As in DINO, target encoder updates are performed with momentum following a cosine schedule between 0.996 and 1. Scaling temperatures are $\tau = 0.1$ and $\tau_t = 0.07$, with the latter being linearly increased from the initial $0.04$ during a 30-epoch warmup stage. Learning rate is linearly ramped up to the base value of 0.0003 during the first 10 epochs and decayed following a cosine schedule. If not mentioned, a hyperparameter has the same value as in DINO. Finally, we set $\lambda_{oc} = \lambda_{global} = 1$.

## 4.2 ANALYSIS

Table 1: FG-ARI and mBO for one synthetic (MOVi-E) and one real-world (EntitySeg) dataset attained by different variants of OCEBO. The first row shows the base OCEBO model trained on COCO, while the subsequent rows represent modifications. Column "Collapse" refers to whether slot collapse has occurred (color green for No denotes that this is a desired scenario). We report results on two datasets for brevity but find the conclusions to be identical across other datasets in the zero-shot benchmark (Didolkar et al., 2024). We report the quantitative measure of slot collapse $d$ described in Section 4.2.

| | Model | d | Collapse | MOVi-E | | EntitySeg | |
|---|---|---|---|---|---|---|---|
| | | | | FG-ARI | mBO | FG-ARI | mBO |
| | OCEBO | 0.13 | NO | 54.8 | 25.8 | 41.5 | 15.3 |
| (a) | W/o patch filtering | 0.02 | YES | 27.7 | 10.6 | 31.7 | 10.0 |
| (b) | $\lambda_{oc} = 0$ | 0.02 | YES | 8.0 | 3.4 | 30.1 | 7.6 |
| (c) | Before sharpening | 0.21 | NO | 44.0 | 20.8 | 39.4 | 12.8 |
| (d) | COCO+ | 0.22 | NO | 66.8 | 22.1 | 44.2 | 16.0 |

**Measuring slot collapse.** Slot collapse is the main mode of failure of object-centric models. When slot collapse occurs, slots do not attach to meaningful subparts of the scene, i.e., objects, but rather to positionally coherent regions of the image (e.g., block-like structures or bands going from top to bottom or left to right). Hence, avoiding slot collapse is of the utmost importance when training object-centric models. The simplest way to identify slot collapse is to qualitatively observe predictions made by the model. In addition, we introduce a quantitative measure of slot collapse. Consider a patch present in both views of the image and assume it is located at index $i$ in view 1 and index $j$ in view 2, i.e., $\boldsymbol{q}_{1,i}$ and $\boldsymbol{q}_{2,j}$. Consider also a patch in view 2 at index $i$, i.e., $\boldsymbol{q}_{2,i}$. Intuitively, when no slot collapse occurs, the similarity of $\boldsymbol{q}_{1,i}$ and $\boldsymbol{q}_{2,j}$ must be higher than that of $\boldsymbol{q}_{1,i}$ and $\boldsymbol{q}_{2,i}$. When a positional collapse occurs, the opposite holds. We introduce a measure $d = \texttt{sim}(\boldsymbol{q}_{1,i}, \boldsymbol{q}_{2,j}) - \texttt{sim}(\boldsymbol{q}_{1,i}, \boldsymbol{q}_{2,i})$ averaged over the validation set as a measure of slot collapse.

**Cross-view patch filtering.** First and foremost, we analyze OCEBO's slot collapse avoidance capabilities. The lack of informative reconstruction targets has been shown to be the main reason

for slot collapse in object-centric models Seitzer et al. (2022). Due to random initialization of our target encoder, reconstruction targets are noisy and not always informative in the initial stages of training. Cross-view patch filtering therefore serves as a collapse prevention mechanism by filtering the patches that should not be considered for self-distillation. As indicated in Table 1 (a), omitting this mechanism immediately results in slot collapse. The importance of patch-filtering is further illustrated by Figure 2. In the first epoch of training, only ∼10% of patches satisfy the patch filtering condition, meaning that the vast majority of reconstruction target is not informative and should not be used. As the training progresses, the percentage of supervised patches increases drastically and finally starts plateauing at ∼70% around epoch 200. We perform additional ablations of the cross-view patch filtering approach in Appendix B.

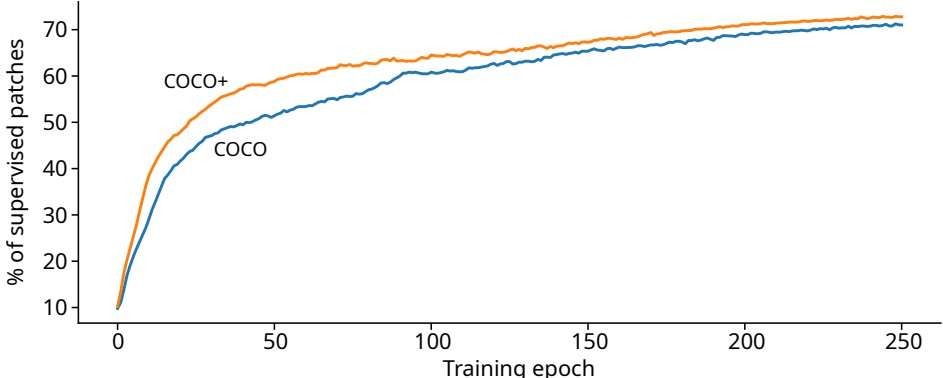

Figure 2: The percentage of supervised patches, i.e., those that satisfy the cross-view patch filtering condition throughout the model training. Blue line corresponds to the model trained on COCO, while the orange line corresponds to that trained on COCO+.

**Object-centric inductive biases.** Second, we demonstrate the importance of injecting object-centric inductive biases into the target encoder by using the object-centric training objective and EMA updates of the target encoder. Table 1 (b) shows the results achieved without the object-centric training objective, i.e., by setting $\lambda_{oc} = 0$ and training slot attention encoder and decoder only during the mask sharpening stage. This reduces OCEBO to pretraining of a DINO model on COCO followed by FT-DINOSAUR fine-tuning with the frozen COCO-pretrained DINO target encoder. Interestingly, this also leads to collapse, although a different mode of collapse in which all pixels are assigned to one slot. Hence, DINO pretraining on COCO is not sufficient to learn informative reconstruction targets.

Moreover, the importance of object-centric inductive biases can be clearly observed by comparing PCA visualizations (see Figure 3) of features produced by OCEBO and DINOv2. In the first three columns, OCEBO separates the bear and human instances, while DINOv2 groups them together. Another interesting observation can be made from the remaining columns – OCEBO learns to encode not only instances but also the part-whole hierarchies. In the first three principal components, the food plate is treated as a whole, while principal components 4–6 encode each food type as a separate instance. Similarly, different groups of people are further divided in principal components 4–6.

**Mask sharpening.** Interestingly, segmentation masks produced by the slot decoder of OCEBO exhibit a lack of clear boundaries between objects. We attribute this to the constant change of reconstruction targets. Introducing a shorter mask sharpening stage in which the targets are kept frozen and the self-distillation objective is replaced by a typical $\ell_2$ loss improves the quality of segmentation masks, as indicated In Table 1(c).

**Scalability.** Object-centric approaches that rely on pretrained encoders have been shown to exhibit poor scalability with added data Didolkar et al. (2024). In fact, when trained on COCO, their performance saturates with a subset of 16384 COCO images. To achieve large-scale pretraining of object-centric models, they must actually scale with data. We demonstrate the scalability of OCEBO by training it on COCO+, a dataset roughly twice the size of COCO. Table 1(d) indicates that the performance increases consistently across datasets in terms of FG-ARI. On MOVi-E, we observe a

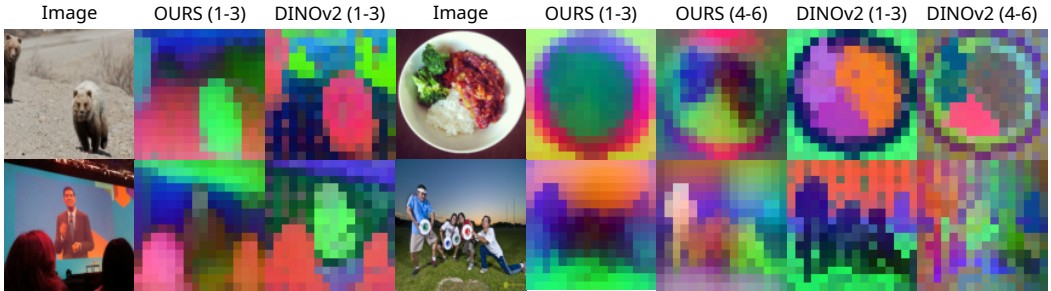

Figure 3: PCA visualizations of the representations produced by the target encoder of OCEBO and by DINOv2. RGB values correspond to principal components 1–3 or 4–6.

slight decrease in mBO but at the cost of high increase in FG-ARI. As we note in more detail in Section 4.3, a trade-off between FG-ARI and mBO should always be taken into account. However, we argue that a large increase in FG-ARI is highly beneficial as simple techniques, such as a short high-resolution training stage (Didolkar et al., 2024) can boost the performance in terms of mBO. We provide a more detailed scaling plot in Appendix C.

However, scale of the data is not the only important consideration. COCO is a curated dataset containing complex scenes with several objects per scene. This is exactly the type of data required by object-centric models. For instance, ImageNet (Russakovsky et al., 2015) is an order of magnitude larger than COCO (∼1.3M images) but is not a suitable pretraining dataset for object-centric models as it mostly contains simple scenes with a single object taking up a large and central part of the scene. In fact, an attempt to train OCEBO on ImageNet results in a drastically lower performance (FG-ARI 46.0, mBO 16.0 on MOVi-E and 39.5 FG-ARI, 11.8 mBO on EntitySeg). Constructing a large-scale dataset suitable for pretraining of object-centric models remains an open question.

## 4.3  COMPARISON TO STATE-OF-THE-ART OBJECT-CENTRIC MODELS

So far, we have demonstrated that OCEBO enables pretraining of object-centric models from scratch on real-world data while avoiding slot collapse. To put its performance in context, we compare it with state-of-the-art object-centric models. The results are reported in Table 2.

Table 2: Performance comparison of OCEBO and state-of-the-art object-centric models. Numbers for models other than OCEBO are taken from Didolkar et al. (2024).

| | Pretrained encoder | MOVi-C | | MOVi-E | | Pascal VOC | | EntitySeg | |
|---|---|---|---|---|---|---|---|---|---|
| | | FG-ARI | mBO | FG-ARI | mBO | FG-ARI | mBO | FG-ARI | mBO |
| SlotDiffusion (Wu et al., 2023) | DINO | 66.9 | 43.6 | 67.6 | 26.4 | 21.1 | 42.0 | 43.7 | 25.1 |
| SPOT (Kakogeorgiou et al., 2024) | DINO | 63.0 | 40.8 | 47.8 | 21.5 | 21.2 | 50.6 | 41.7 | 27.4 |
| DINOSAUR (Seitzer et al., 2022) | DINOv2 | 67.0 | 34.5 | 71.1 | 24.2 | 24.0 | 37.2 | 43.5 | 19.4 |
| FT-DINOSAUR (Didolkar et al., 2024) | DINOv2 | 71.3 | 44.2 | 71.1 | 29.9 | 24.0 | 37.6 | 48.1 | 28.4 |
| OCEBO | None | 63.1 | 27.3 | 66.8 | 22.1 | 22.4 | 34.4 | 44.2 | 16.0 |

We note several things here. First, the models are not directly comparable. OCEBO is pretrained from scratch on COCO+, while the rest of the models rely on image encoders pretrained on 1.3M (DINO) or 142M (DINOv2) images. Second, in this work we do not focus on tuning the performance and achieving the highest possible numbers. For instance, the autoregressive decoding strategy of SPOT Kakogeorgiou et al. (2024) is known to significantly improve the mBO metric, while FT-DINOSAUR (Didolkar et al., 2024) uses a top-k MLP decoder and a short high-resolution training stage that improve both FG-ARI and mBO. Rather than incorporating these additional components, we aim to demonstrate that pretraining is possible and scales well with data. Moreover, the results indicate that we can even achieve performance comparable to that of state-of-the-art models, although OCEBO has seen orders of magnitude less images during training. This emphasizes the importance of object-centric pretraining rather than adapting an encoder already pretrained in

a non-object-centric way. However, we also predict that incorporating components from SPOT or FT-DINOSAUR (or subsequent works) will be of the utmost importance for reaching the highest possible performance. Finally, there is always a trade-off between FG-ARI and mBO inherently present in every model. For instance, models such as DINOSAUR Seitzer et al. (2022) attain higher FG-ARI at the cost of mBO due to the use of a MLP-based decoder, which we also use in OCEBO (we observe the same trend here). On the other hand, autoregressive decoders such as that of SPOT (Kakogeorgiou et al., 2024) increase mBO at the cost of FG-ARI. Therefore, comparing performance gains is further complicated by this as no metric is clearly better than others. This can be clearly observed by comparing methods in Table 2.

## 5 CONCLUSION

We propose OCEBO, the first ever pretraining scheme for object-centric models. The inspiration for this work stems from the limitations of state-of-the-art object-centric models caused by the use of frozen pretrained target encoders that cannot be meaningfully updated during training and thus impose an upper bound on attainable performance and result in poor data scalability. We demonstrate that through EMA updates of the target encoder we remove the upper limit and that training from scratch allows the target encoder to capture object-centric inductive biases, thus utilizing the data to its full extent. OCEBO scales well with dataset size and achieves comparable performance despite seeing only a fraction of training data seen by pretrained non-object-centric models used as target encoders in other works.

A notable limitation of this work is the dataset size used for pretraining. Namely, for object-centric pretraining is an open question as datasets with simple scenes such as ImageNet prevent object-centric models from capturing the most important information – objects. We hope that this work and the novel insights it brings (most importantly, the possibility to scalably pretrain object-centric models) inspires the community to explore object-centric pretraining on larger datasets and ultimately pursue the goal of large-scale object-centric foundation models.

**Reproducibility Statement.** Code and models will be made available upon acceptance of the paper in a public repository. This will include a README file with instructions for setting up the environment and reproducing the experiments All datasets used are publicly available.

**Acknowledgements.** This research was supported by funding from the Flemish Government under the "Onderzoeksprogramma Artificiele Intelligentie (AI) Vlaanderen". This work was also partially funded by the European Research Council (ERC) under the European Union's Horizon 2020 research and innovation program (Grant Agreement No. 101021347). We acknowledge EuroCC Belgium for awarding this project access to the LUMI supercomputer, owned by the EuroHPC Joint Undertaking, hosted by CSC (Finland) and the LUMI consortium. The resources and services used in this work were provided by the VSC (Flemish Supercomputer Center), funded by the Research Foundation - Flanders (FWO) and the Flemish Government.

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

## A    ESTIMATING THE QUALITY OF PATCH-LEVEL REPRESENTATIONS

Object-centric models that keep the pretrained encoder frozen throughout the training inherently keep the encoder as is and retain its patch-level representation capabilities. On the other hand, works like SPOT or FT-DINOSAUR fine-tune the encoder and thus change its weights and the quality of patch-level representations. Moreover, as OCEBO is pretrained from scratch, it would ideally learn not only good object-level representations but also good patch-level representation, thus becoming a versatile model useful for a variety of downstream tasks. To verify this, we evaluate object-centric models in a dense downstream task of in-context semantic segmentation (i.e., retrieval-based scene understanding), as described in Balaževic et al. (2023); Lebailly et al. (2024). We evaluate on Pascal VOC Everingham et al. (2010), subsampling its training set with factors of 128, 64, 8 and 1. We fit a nearest neighbors classifier with the number of neighbors set to 50 and evaluate the performance on the full validation set. We compare OCEBO to a state-of-the-art patch-level self-supervised approach CrOC (Stegmüller et al., 2023). For SPOT and FT-DINOSAUR, we report the performance of their original backbones and the fine-tuned versions. The results are reported in Table 3. OCEBO achieves performance comparable to that achieved by state-of-the-art patch-level model. Interestingly, fine-tuning the encoder for object-centric learning degrades the quality of patch representations, although not drastically. This is expected as a consequence of non-object-centric model serving as a target encoder, thus forcing the object-centric model to adjust its features towards slightly more spatially-oriented representations.

Table 3: Retrieval-based scene understanding performance. We report mIOU on the Pascal VOC dataset, where the training set is subsampled with factors of 128, 64, 8 and 1.

| Model | Backbone architecture | Pretraining dataset | 1/128 | 1/64 | 1/8 | 1/1 |
|---|---|---|---|---|---|---|
| OCEBO | ViT-S/16 | COCO | 28.5 | 32.3 | 39.6 | 46.2 |
| CrOC | ViT-S/16 | COCO | 27.1 | 31.4 | 40.3 | 47.1 |
| SPOT | ViT-B/16 | ImageNet | 25.9 | 31.9 | 41.9 | 50.2 |
| DINO | ViT-B/16 | ImageNet | 29.2 | 34.7 | 47.2 | 54.9 |
| FT-DINOSAUR | ViT-S/14 | LVD142M | 33.9 | 43.1 | 58.1 | 65.7 |
| DINOv2 | ViT-S/14 | LVD142M | 46.9 | 53.7 | 64.7 | 69.05 |

## B    ADDITIONAL ABLATIONS OF CROSS-VIEW PATCH FILTERING

In Section 4.2 we demonstrate that without the cross-view patch filtering strategy, OCEBO suffers from slot collapse. However, one could wonder whether the cross-view strategy is necessary or if a simpler heuristic would achieve the same effect. We design two simpler heuristics that rely on 1) supervising all patches but giving the object-centric objective low weight and increasing it throughout the training or 2) randomly selecting patches to supervise with the drop ratio decreasing throughout the training. The results are shown in Table 4. Although better than the case where no object-centric objective is used (see Table 1 (b)), both versions fall short to the cross-view patch filtering strategy of OCEBO. Interestingly, the slot collapse measure is significantly lower compared to OCEBO and qualitative inspection shows predictions not completely collapsed but closer to collapse in comparison with OCEBO. This further corroborates the need for selecting informative patches rather than just letting the global loss drive initial stages of training.

Table 4: FG-ARI and mBO for one synthetic (MOVi-E) and one real-world (EntitySeg) dataset attained by different variants of OCEBO's cross-view patch filtering approach. The first row shows the base OCEBO model trained on COCO, while the subsequent rows represent modifications. Column "Collapse" refers to whether slot collapse has occurred (color green for No denotes that this is a desired scenario). We report results on two datasets for brevity but find the conclusions to be identical across other datasets in the zero-shot benchmark (Didolkar et al., 2024). We report the quantitative measure of slot collapse $d$ described in Section 4.2.

|   | Model | d | Collapse | MOVi-E | | EntitySeg | |
|---|-------|---|----------|--------|---|-----------|---|
|   |       |   |          | FG-ARI | mBO | FG-ARI | mBO |
|   | OCEBO | 0.093 | NO | 54.8 | 25.8 | 41.5 | 15.3 |
| (a) | All patches, increasing weight | -0.017 | ? | 49.2 | 22.4 | 36.9 | 13.5 |
| (b) | Random patches, decreasing drop ratio | -0.026 | ? | 44.0 | 20.0 | 36.7 | 12.7 |

## C  SCALING PLOTS

From Section 4.2 we determine that OCEBO scales well in the range of COCO and COCO+ dataset sizes. In Figure 4 we aim to better depict the scaling laws by providing a few more data points. The plots indicate that the model still scales well and does not saturate near the size of COCO+ dataset, hinting that further scaling is possible. Note that due to the design of OCEBO (MLP decoder of slot attention), it is among the object-centric methods that favor FG-ARI rather than mBO, as discussed in Section 4.3.

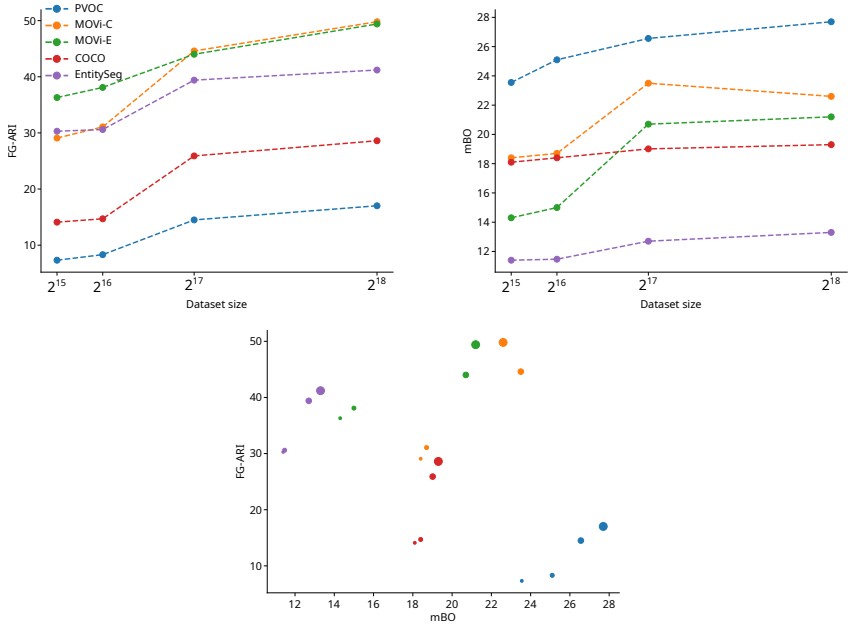

Figure 4: Top: Scaling plots for FG-ARI (left) and mBO (right) with dataset sizes of $2^{15}$, $2^{16}$, $2^{17}$ and $2^{18}$ sampled from COCO or COCO+. Bottom: FG-ARI vs. mBO plot where point sizes indicate the dataset size (the smallest point corresponds to $2^{15}$, while the largest corresponds to $2^{18}$.

