# OpenReview forum: "Object-Centric Pretraining via Target Encoder Bootstrapping"
_ICLR.cc/2025/Conference — ICLR 2025 Poster_

### Official Review · Reviewer_pa7a · 2024-10-28

**Soundness:** 2
**Presentation:** 3
**Contribution:** 2
**Rating:** 6
**Confidence:** 4

**Summary:**

This paper studies the problem of effectively updating the target encoder in object-centric pre-training. Previous works use frozen pre-trained encoders as the target encoder, resulting in a performance upper limit. While updating the pre-trained encoders causes a significant performance drop, the paper proposes to bootstrap the target encoder from scratch. To prevent slot collapse, a cross-view patch filtering technique is proposed. Experiments show that OCEBO can be trained from scratch and learn from more data.

**Strengths:**

1. The paper is easy to follow and understand.
2. The motivations for cross-view patch filtering and mask sharpening stage are straight-forward and these two techniques are proven to be effective.

**Weaknesses:**

1. The experimental evidence for scalability is too weak. A scaling plot, which shows how the model performs as training data increases, is more supportive. From only two data points, it's hard to tell the scaling trend. For example, what if the model is just in rapid growth on 100k images and has already plateaued on 200k images? The authors are suggested to provide a scaling plot instead of two data points.
2. There still seem to be large gaps between the final results and previous methods, which can not support the claim that OCEBO is comparable to those with pre-trained encoders.
3. Discussion on object-centric data and non-object-centric data should be added. While a frozen target encoder can be an upper limit, a feasible way is to use stronger target encoders, as shown in the comparison between DINOv2 and DINO. Stronger DINO can be trained using more data, where object-centric data and non-object-centric data can both be used. So what's the benefit of scaling object-centric data over scaling pre-trained data for target encoders?

**Questions:**

1. The authors are suggested to provide more evidence on scalability. Moreover, it would be better to provide an estimation of data amount required to achieve comparable performance with SOTA models.
2. More benchmarks should be compared. This paper only reported MOVi-C, MOVi-E, Pascal VOC, EntitySeg results, while only two of them are real-world datasets. The authors are suggested to add more real-world datasets, especially the COCO dataset.

---

> ### Author Response · Authors · 2024-11-20
> **Response to reviewer pa7a**
>
> Dear reviewer, thank you for a constructive review and for recognizing the clear motivation behind some of the main contributions of OCEBO. Below we attempt to resolve your concerns and answer your questions and would be more than happy to discuss anything in detail.
>
> **More evidence on scalability (W1 and Q1)**
>
> Thank you for a great suggestion. We include a scaling plot in Appendix C. It suggests that the performance does not plateau yet and that scaling up is well worth exploring. We think that making a claim as to how much data would be needed to achieve sota performance might be risky, especially as the final numbers are not really directly comparable (we elaborate further in the following answer).
>
> **Comparison to sota (W2)**
>
> As OCEBO is the first object-centric method successfully applied to real-world data without relying on backbones pretrained on millions of images, comparing its performance to current sota approaches has been a challenge. Our paper’s focus is to relay a message that pretraininig from scratch is possible, which has been believed not to be the case by now. To reduce the noise in the paper, we intentionally refrain from introducing additional tricks into OCEBO that could easily improve the performance and reduce the current gap in numbers (e.g., the autoregressive decoding strategy of SPOT or the short high-resolution training stage of FT-DINOSAUR). Moreover, the performance gap is to be expected at the current stage due to the drastically lower amount of data that OCEBO relies on. However, all the evidence (e.g., entries (b) and (d) in Table 1 and the scaling plot in Appendix C) suggests that scaling up is possible and might quickly close the gap between OCEBO and current sota. That being said, our computational resources are quite limited and quickly verifying this is not easy. This is the main reason we believe OCEBO should be released into the community, thus inspiring others to pursue the paradigm of object-centric pretraining.
>
> **Better encoders and object-centric vs. non-object-centric data (W3)**
>
> As you correctly note, curated datasets with simple scenes such as ImageNet are not suitable for object-centric pretraining. However, we believe that obtaining sufficient amounts of suitable data is not a large hurdle. In fact, any uncurated dataset such as LAION[1] or Open Images Dataset[2] contains complex scenes with several objects per image and could in our opinion serve as a suitable off-the-shelf alternative.
>
> Of course, as you note, another option is to train increasingly powerful non-object-centric backbones that have been successfully trained on datasets such as ImageNet. However, we think that there might be diminishing returns from this, especially given that the current sota models such as DINOv2 no longer rely on curated data such as ImageNet. Scaling further than DINOv2 is a quite gruesome task requiring enormous amounts of data and computational resources. Moreover, from the scaling laws and our initial experiments (see entry (b) in Table 1), it seems that with the same amout of data, directly pretraining object-centric models is more beneficial than relying on non-object-centric backbones.
>
> With all this, we believe that object-centric pretraining is a promising direction and could bring a paradigm shift to object-centric learning and representation learning in general, where it could lead to the emergence of generic global-patch-object level backbones.
>
> [1] Schuhmann et al.: LAION-400M: Open Dataset of CLIP-Filtered 400 Million Image-Text Pairs. 2021
>
> [2] Kuznetsova et al.: The Open Images Dataset V4: Unified image classification, object detection, and visual relationship detection at scale. 2020.
>
> **More benchmarks (Q2)**
>
> We agree with your remark that more benchmarks, especially real-world ones could be beneficial. However, to avoid diluting the main message of OCEBO, we decided to follow the standard evaluation protocols introduced by the previous sota object-centric approaches. Until recently, the trend has been to train and evaluate object-centric models on each dataset separately and the span of datasets has been limited to COCO, Pascal VOC and MOVi datasets. FT-DINOSAUR[3] introduced a new framework where we train on COCO and evaluate on other datasets and added a few more datasets to the benchmark (e.g., EntitySeg). We decided to follow this protocol for the best possible comparison (although it is far from perfect as we note in previous responses). We refrain from using COCO as an evaluation dataset as it is already used as a pretraining dataset according to FT-DINOSAUR. Evaluation protocols for object-centric models is in our opinion a very important open question that deserves more attention, but we believe this to be out of the scope of our current work.
>
> [3] Didolkar et al.: Zero-Shot Object-Centric Representation Learning. 2024

---

> > ### Comment · Reviewer_pa7a · 2024-11-25
> > **Response to Authors**
> >
> > Thanks for the authors' response. I have read the authors' responses and reviews of other reviewers. However, my major concern still remains:
> > 1. Training data scalability. Figure4 in the appendix is not complete. 2^17 dataset size is only 131k, but the model actually has been trained on 241k images, why not add this point to the plot? Furthermore, the authors are suggested to plot the absolute performance, including FG-ARI and mBO metrics, rather than relative performance. This can provide a more comprehensive description of data scaling.
> > 2. Comparison with other models. I understand that computational resources can be a bottleneck, but the paper is responsible for demonstrating clear messages. However, the current comparison looks confusing: FG-ARI and mBO present different conclusions, and, as the authors noted, some methods adopt techniques for higher FG-ARI, while some other methods favor improving mBO. It's really hard to tell if the performance gap comes from the proposed method or existing tricks. The authors are suggested to reorganize the comparison to show the real impact of OCEBO. For example, list all the influence factors in the table and use checks to clearly indicate which technique has been adopted by each method. Moreover, it would be better if the authors could apply these techniques to OCEBO, like the autoregressive decoding strategy of SPOT or the short high-resolution training stage of FT-DINOSAUR. From my perspective, this does not complicate the conclusion, but provides a more clear comparison to show the effectiveness of the proposed method.

---

> ### Author Response · Authors · 2024-11-26
> **Response to reviewer pa7a**
>
> We are sorry to hear the main concerns remain.
>
> **1.**
>
> Regarding the first remark, i.e., not including the 241k data point, this is an unfortunate labeling mistake. There seems to have been a rounding error in visualization code that resulted in an error by factor 2 in x axis labels. In the previous revision of the manuscript, we already discussed Figure 4 as if the COCO+ results were present, and indeed they were. We apologize for this inconvenience and fix the labels in the manuscript.
>
>    Initially, as Figure 4 contains multiple datasets, we opted for relative metrics for easier comprehension, i.e., in terms of performance gain. Nonetheless, we replace the relative FG-ARI with the absolute and add mBO in the revised manuscript, as suggested. In addition, we include a third plot of FG-ARI vs mBO, which we believe is the most informative. Conclusions remain the same: OCEBO indeed still scales at the COCO+ size. As noted, this is not as clear from the mBO plot but this is due to the known trade-off between FG-ARI and mBO. As long as one metric keeps scaling, there is space to improve the other by balancing the trade-off (more on this in the continuation of this response).
>
> **2.**
>
> We understand the reasoning behind reviewer's concerns but we feel there's a disagreement between our views of the main message of OCEBO. So far, all sota object-centric works have relied on freezing a pretrained backbone and using its features as reconstruction targets. The main difference between methods lied in adding components that help the training in one way or another. If OCEBO was another work following the same paradigm, we definitely agree that presenting experimental results (i.e., the numbers) in a fair and reliable way would be necessary to ensure a clear message of the paper. In that case, OCEBO would be directly comparable to DINOSAUR, SPOT, etc.
>
> On the other hand, OCEBO paves a way to a new paradigm: one where we pretrain object-centric models from scratch without relying on the backbones pretrained in a non-object-centric manner. As demonstrated, this paradigm is scalable and the contributions of OCEBO allow it to avoid slot collapse while achieving favorable backbone properties (e.g., the separation of instances and capturing hierarchies). We see this as the main message of our work. Of course, the question of scaling up further than COCO+ still remains (and hence the computational limitations argument), but we argue that this is a hurdle that can be overcome by the community and does not harm the clarity of the main message.
>
> To further motivate OCEBO, we report the results on a typical task used to evaluate object-centric models in Table 2. We demonstrate that with orders of magnitude less data (SlotDiffusion and SPOT use DINO backbone pretrained on 1.3M images + COCO, while DINOSAUR and FT-DINOSAUR use DINOv2 pretrained on 142M images + COCO), OCEBO avoids slot collapse and comes satisfyingly close to sota methods in terms of performance. We have thought long and hard how best to compare with other methods. One option would indeed be to go method by method, add every component we can from that method to OCEBO and compare the results this way (pairwise). Another option would be to remove some components (e.g., the autoregressive decoding, high-res training stage or replacing DINOv2 backbone with DINO) from corresponding models and compare that way. We believe that either would reduce the performance gap. However, the reality is that the performance gap would still be present due to the difference in training data (we explicitly emphasize this difference by mentioning the backbones in Table 2). In addition, it is important to be aware that the gap stems from several factors that can hardly be disentangled: perhaps some of the tricks work better with different training strategies, etc. This is what prompted us to keep the simplest version of OCEBO and best performing versions of other methods. In our opinion, this does not impact the main message that pretraining is possible and that efforts should be made to see how far we can go in this direction.
>
> Finally, we would like to address the argument that FG-ARI and mBO present different conclusions. If we disregard OCEBO in Table 2 and just compare other sota methods, we can still observe a quite significant trade-off between methods that favor FG-ARI (DINOSAUR and FT-DINOSAUR) and those that favor mBO (SlotDiffusion and SPOT). The evaluation of object-centric models is in our opinion flawed and deserves more attention in the form of works dedicated exclusively to this problem, but we believe this should not be held against the work we present here. We do our best to demonstrate OCEBO's "performance" within the framework widely adopted at the moment, but still believe the actual numbers to be less relevant than the main message.
>
> &nbsp;
> We are looking forward to hearing the reviewer's views on this.

---

> > ### Comment · Reviewer_pa7a · 2024-11-27
> > **Response to Authors**
> >
> > Thanks for the authors' response. I have no further questions about the scaling plot. Regarding the performance comparison, I agree that the primary gap comes from pre-training data size. I recommend the authors consider pre-training with a larger dataset, such as LAION, and include the results in the paper, which I believe will make the paper more complete. I will increase my score to 6.

---

### Official Review · Reviewer_hsmN · 2024-11-01

**Soundness:** 3
**Presentation:** 2
**Contribution:** 2
**Rating:** 6
**Confidence:** 3

**Summary:**

This work proposed an object-centric pretraining method that updates the target encoder by EMA. The experiment results show that the proposed method can successfully learn object-centric representation. When pretrained on 241k images from COCO, the proposed achieves unsupervised object discovery performance comparable to other models with frozen non-object-centric target encoders pretrained on hundreds of millions of images.

**Strengths:**

1. This paper is well written and easy to follow.
2. The proposed method can achieve unsupervised object discovery performance comparable to other models with frozen non-object-centric target encoders pretrained on hundreds of millions of images.
3. The proposed method demonstrates scalability well beyond a few thousand training images.

**Weaknesses:**

1. How exactly object-centric inductive biases are captured by the target encoder, it may be better to explain the mechanism more intuitively or theoretically.
2. As the author mentioned, although the proposed method has achieved comparable results in COCO pre-training, its advantage still needs to be verified on a larger scale of pre-training data.

**Questions:**

1. What distance is used when calculating nearest neighbors?
2. I don't understand what is the meaning of ``invaug(q1)''.

---

> ### Author Response · Authors · 2024-11-20
> **Response to reviewer hsmN**
>
> Dear reviewer, thank you for a positive review and for recognizing the strengths of our work. Below we address your concerns and questions. We are happy to discuss anything else at more detail.
>
> **How are the object-centric inductive biases captured by the target encoder (W1)?**
>
> As the object-centric model contains slot attention encoder and decoder, object-centric inductive biases are propagated back to the encoder (backbone) in terms of gradients. Since the target encoder’s weights are updated as an exponential moving average of the object-centric model’s encoder, it gradually gets enriched by same inductive biases accumulated in the object-centric model.
>
> In contrast, previous works such as DINOSAUR or SPOT rely on a target encoder that is pretrained in a non-object-centric way (e.g., DINO or DINOv2) and frozen during training, which means that it never receives any object-centric inductive biases. We will try to make this more intuitive in the paper.
>
> **Large-scale pretraining still needs to be performed**
>
> Indeed, large-scale pretraining is the natural next step for OCEBO. Due to computational constraints, we have not been able to perform this pretraining yet, but we believe this to be all the more reason to introduce OCEBO to the community, thus inspiring others to pursue the same goal. As mentioned in the paper, pretraining on simple scenes from ImageNet is not a sufficient signal for object-centric pretraining but we believe that any uncurated dataset (e.g.,  LAION[1] or Open Image Dataset[2]) contains a sufficient number of complex scenes with several (potentially interacting) objects to enable large-scale object-centric pretraining. Moreover, we would like to note that the achieved numbers are not directly comparable with sota methods, and more importantly that they are not as central as the main message of OCEBO, which is that pretraining from scratch is indeed possible and that it might be well worth exploring.
>
> [1] Schuhmann et al.: LAION-400M: Open Dataset of CLIP-Filtered 400 Million Image-Text Pairs. 2021
>
> [2] Kuznetsova et al.: The Open Images Dataset V4: Unified image classification, object detection, and visual relationship detection at scale. 2020.
>
> **Distance metric for nearest neighbors**
>
> We use the cosine distance between patch representations to determine the nearest neighbors of each patch. Of course, there might be a more suitable distance metric, but cosine is common and seemingly sufficient in other self-supervised works.
>
> **What is invaug(q1)?**
>
> The operation invaug refers to inverse augmentation. To obtain two views of an input image, we apply two sets of data augmentations. However, due to random cropping and horizontal flipping of the image, a pixel with index 1_1 in one view will not be located on index 1_1 in the other view. This is where inverse augmentation comes into play. Basically, what it does is it finds parts of the input image present in both views, cuts them from each view and resizes the cut regions back into some common size.
>
> We can do the same operation on features rather than the input image. When we obtain features q1 and q2 from both views, we need to perform the operation of inverse augmentation to ensure they are aligned, i.e., that the features q1 at index 1_1 correspond to features q2 at index 1_1 and so on. In this case, invaug will use the parameters of applied data augmentations to align the features in the desired way. For a graphical representation of inverse augmentation, we refer to Figure 2 of SlotCon [1].
>
> [1] Wen et al.: Self-supervised visual representation learning with semantic grouping. 2022

---

> > ### Comment · Reviewer_hsmN · 2024-11-27
> >
> > Thanks for the author's response. I maintain a positive rating for this paper.

---

### Official Review · Reviewer_a9mq · 2024-11-03

**Soundness:** 2
**Presentation:** 3
**Contribution:** 3
**Rating:** 8
**Confidence:** 5

**Summary:**

The authors propose an approach to train object-centric models from scratch using real-world data, rather than relying on pre-trained non-object-centric foundation models.
The method is based on cross-view teacher-student self-distillation, in a similar fashion to DINO, IBOT and DINOv2.
The model architecture incorporates a slot-attention bottleneck and the patch-level loss uses a filtering strategy to stabilize training.
The method is trained on COCO and evaluated on different datasets on the task of unsupervised object discovery, where it attains performance comparable to (but lower than) previous methods that leverage large-scale pre-trained models.

**Strengths:**

The main strength of the paper is succeeding in training an object-centric model from scratch on COCO, which is known from previous works to be challenging.
The architecture or training procedure per-se are not particularly novel, mostly resembling the global and patch losses of DINO, IBOT and DINOv2, with the addition of a slot-attention bottleneck in the architecture.

What is novel is the idea of filtering noisy patches that could be detrimental to the object-centric objective, especially during the first stages of training.
This idea, albeit not well ablated, seems to be a strong contribution of the paper.

The paper is easy to follow, with a good balance between technical details, analogies, and high-level explanations.
Quantitative results are presented clearly and accompanied by qualitative examples.

**Weaknesses:**

**Ablation on patch filtering:**
From section 4.2, it appears that patch filtering is crucial to stabilize training.
The chosen strategy uses an heuristic to filter out patches, especially during the first stages of training, as show in figure 2.
The first question that comes to mind is: how sensitive is the method to the choice of the heuristic?
It could be that the chosen heuristic has no importance and what really matters is that initially the global loss drives the training and the object loss is introduced gradually later.
In my opinion, this is an important ablation study to perform in the paper.
Two alternatives that I would like to see tested are:
- Keeping all the patches but gradually increasing $ \lambda_{oc} $ from 0 to 1 during training.
- Randomly dropping patches in $ \mathcal{L}_{oc} $ as opposed to selecting them via nearest neighbors. The drop ratio could be gradually increased from 0 to 1 during training to mimic the proposed heuristic.

**Measuring slot collapse:**
An important point of discussion is "slot collapse", defined in the footnote at L107.
Since the authors claim that the proposed patch filtering strategy is crucial to avoid slot collapse, it would be helpful to have a quantitative and objective metric to measure slot collapse.
This could be, for example, the correlation between slots and spatial positions across images, to measure whether a slot encodes the "bottom right corner" or a category of objects.
The green/red results in table 1 would be more informative and convincing if accompanied by such a metric.

**One model or several ones?**
The whole model is trained from scratch on COCO and evaluated on different datasets, each with a specific number of slots (L319).
Does it mean that a new model needs to be trained from scratch for each number of slots?
If so, this is highly impractical for real-world applications where a practitioner would like to sweep over the number of slots to find the best one.
In such a case, frameworks like DINOSAUR or SPOT are much less expensive to use.
If not, how is the number of slots changed in the model? Is it fixed before training or can it be changed at inference?

**No evaluation of the learned representation:**
All evaluations focus on segmentation-based metrics (FG-ARI and mBO) on several datasets.
The task of "object-centric learning", however, implies that the model should learn a representation of objects, not just segment them.
It would be useful to include a section that evaluates the slot representation on downstream tasks in a quantitative manner.

**Projection head design:**
On L328-331 it says "The projection heads are identical to those of DINO (Caron et al., 2021), with the exception of setting L = 8192 instead of the original 65536. Compared to the DINO head, ours projects every patch rather than just the global representations and we find that the gain in performance does not justify the computational cost."
However, both IBOT and DINOv2 use per-patch heads and find that a large number of heads, even up to 131072, is beneficial.
If time allows, I recommend running an ablation study on the design of the projection heads, possibly splitting the object and global heads.

**Performance and usefulness:**
Weaker performance when compared to other methods that leverage large-scale pre-trained models (table 2).
This is somewhat expected, since the model is trained from scratch on a smaller dataset.
At a high level, this paper demonstrates that training from scratch is possible, but fails to prove that is actually beneficial.
If a pre-trained model achieves better performance, why should one train from scratch?

**Questions:**

**Equation 1:**
I suggest renaming $\mathcal{L}_{oc}$ to something else to avoid confusion with the actual loss used during training which is defined in equation 3.

**Ablation of head design:**
Equations 3 and 4, as well as the filtered version in 8, describe a cross-view teacher-student distillation loss.
This setup requires quite a few moving parts, especially the cropping strategy with overlapping parts and the inverse augmentation.
Would it be possible to train the model without cross-view distillation, but only using the teacher's output on the same crop as the target?

**Comparison with the DINO objective?**
The global loss in equations 5 and 6 is formulated exactly as in DINO, why does the paragraph above it cite other papers and not DINO?

**Suggestion about notation:**
Paragraph 3.3 and line 269 "where $nns_{n_n}(z_{t,1}, z_{t,2})_i$ denotes indices of nn nearest neighbors".

There are too many "n" characters in the chosen notation and it's hard to read.
I suggest trying to replace $n_n$ with $k$ if possible.

**Where is SPOT in the introduction?**
To the best of my knowledge, SPOT is the first work that unfreezes the encoder during training, and it was published months before FT-DINOSAUR.
However, in the introduction, FT-DINOSAUR is presented as the first and is discusses in depth, while SPOT is not mentioned. This is misleading and should be corrected.

**Missing results:**
L435 "In fact, an attempt to train OCEBO on ImageNet results in a drastically lower performance." where are these results?

---

> ### Author Response · Authors · 2024-11-20
> **Response to reviewer a9mq (1/2)**
>
> Dear reviewer, thank you for a thorough positive review and constructive suggestions. We are particularly glad you identify patch filtering as a strong contribution. We hope the answers below resolve your concerns and we are happy to discuss more at any point.
>
> **Ablation on patch filtering**
>
> We wondered about the same thing during the method development. Usually, one can design several heuristics that achieve the same thing, but we believe the central part of their design needs to lie in selecting exactly which patches to reconstruct, rather than reconstructing all (or randomly selected ones) with a lower weight than that of the global loss. If we trained a model with global loss only (i.e., DINO), some patches would be easier for model to understand, while others would be more difficult. If at any stage of the training we force slot attention to reconstruct the latter (noisy) patches, the model could go towards degenerate solutions. OCEBO’s patch filtering method avoids just that by actually filtering out noisy patches and reconstructing only those that the model already understands well. The ablations you propose perfectly support this argument (thank you for suggesting those), as indicated in Appendix B.
>
> **Measuring slot collapse**
>
> This is a great suggestion! We introduce a metric that utilizes positive and negative patch pairs from two views of the image. A more formal introduction and updated ablation table can be found in the revised manuscript, Section 4.2 (L361). The numbers indeed support our claims and make them stronger.
>
> **One model or several ones**
>
> No, there is no need to train a new model for every slot number. We follow the exact same framework as SPOT or DINOSAUR (or other slot attention-based methods): we initialize slots from a learnable distribution and send them through the slot encoder and decoder. We train on COCO with 7 slots per image and can sample an arbitrary number of initial slots at inference time.
>
> **No evaluation of the learned representations**
>
> We completely agree with your argument. Evaluating only on the task of unsupervised object discovery is a current trend in the object-centric literature. We, too, believe that this needs to be challenged by introducing metrics focusing on representation quality. However, this requires the design of novel downstream tasks, which we believe is out of the scope of this work. Here, we aim to provide evaluations within the current standard framework rather than challenge it, although we do plan to do so in our future work.
>
> **Projection head design**
>
> From our understanding of iBOT and DINOv2 projection heads, their design slightly diverges. In iBOT, a projection head of dimension 8192 with shared weights is used for all model configurations. Moreover, section “Output dimension” in Appendix E (second half of page 20) of iBOT suggests that increasing the dimension to 16384 does not improve the performance. We observe the same effect in OCEBO: increasing the head size from 8192 to 65536 with a ViT-S/16 model brings no performance improvements but increases the computation time by 20%.
>
> On the other hand, DINOv2 authors find that, as opposed to iBOT, splitting the heads and increasing the dimension helps. They hypothesize that this different behavior occurs due to scale (top of page 6). As such, we assume that a similar effect could be observed with OCEBO, but we are not there yet as we still don’t train large models on hundreds of millions of images. With such large models, increasing the head size will bring a negligible increase in computation time but seems to improve the performance.
>
> If you are not convinced and the computational resources allow, we would gladly perform a more detailed ablation of the head design.

---

> > ### Author Response · Authors · 2024-11-20
> > **Response to reviewer a9mq (2/2)**
> >
> > **Performance and usefulness**
> >
> > The current paradigm of using pretrained non-object-centric models as frozen target encoders has definitely shown great improvements in terms of unsupervised object discovery metrics. However, the inability to subsequently improve the target encoder imposes an upper limit on the performance and one that can be achieved quite easily.
> >
> > To overcome this performance barrier, one can either push it higher by training larger and more powerful non-object-centric backbones (which requires huge amounts of data and vast computational resources) or find a way to overcome it. The main purpose of OCEBO is to demonstrate that the latter might be possible by pretraining from scratch, which as you note yourself, has been believed to be impossible (or extremely difficult). The current scaling trends we observe (e.g., entries (b) and (d) in Table 1) suggest that scaling up pretrained object-centric models might quickly surpass the performance of approaches relying on non-object-centric backbones.
> >
> > Moreover, we’d like to note that several improvements are possible on top of OCEBO, such as the autoregressive decoding strategy from SPOT or the short high-resolution training stage from FT-DINOSAUR that might easily bring the numbers presented in this work closer to sota. That being said, we do not consider this to be crucial and rather focus on communicating a message that exploring this new direction in object-centric learning could be beneficial. Of course, this remains to be seen but that is the purpose of research.
> >
> > As far as scaling up goes, we think that uncurated datasets such as LAION[1] or Open Image Dataset[2] contain enough complex scenes to allow object-centric pretraining without the need to construct novel datasets. Our current computational capabilities prevent us from verifying this quickly but this is yet another reason why we wanted to share OCEBO with the community, hoping to inspire others to pursue this direction as well.
> >
> > All being said, we believe that the paradigm of object-centric pretraining is well worth exploring and hope it could lead to unified global-patch-object level backbones and that the performance-wise benefits of OCEBO will become more obvious in the long run as we start scaling up and reaping all the benefits of object-centric inductive biases.
> >
> >
> > [1] Schuhmann et al.: LAION-400M: Open Dataset of CLIP-Filtered 400 Million Image-Text Pairs. 2021
> >
> > [2] Kuznetsova et al.: The Open Images Dataset V4: Unified image classification, object detection, and visual relationship detection at scale. 2020.
> >
> > **Notation suggestions**
> >
> > We rename $\mathcal{L}_{oc}$ in equation 1 and the nearest neighbor notations as you suggested. It is indeed more clear now.
> >
> > **Ablation of head design**
> >
> > As argued in the “Ablation on patch filtering” response, we believe that selecting the right patches is crucial for successfully training object-centric models. We rely on cross-view information to determine which patches to select, so completely removing the cross-view strategy would not be possible. What could be possible (and we suspect you refer to) would be to perform distillation from features of the same view while still using cross-view information to select patches, i.e., using equation 1 instead of equation 3. This setting would not drastically simplify the overall method but would remove augmentation invariance from the slot attention module, which we found doesn't directly impact the final performance but could in our opinion be useful for improving patch-level representation quality (we will try to ablate this before the discussion period ends).
> >
> > **Comparison with the DINO objective**
> >
> > You are absolutely right. In the mentioned paragraph we refer to the fact that other patch-level self-distillation works use global loss but fail to mention that it originates from DINO. We added the missing reference.
> >
> > **Where is SPOT in the introduction?**
> >
> > Thank you for catching this oversight. SPOT is indeed the first object-centric model to successfully unfreeze the encoder. Although, in our interpretation, SPOT’s major strengths and main contributions lie elsewhere (autoregressive decoding with permuted sequences and attention self-distillation (regardless of the backbone update)) and that is the reason we failed to mention its fine-tuning together with FT-DINOSAUR. Regardless, we absolutely agree with you and rectify this in the updated introduction.
> >
> > **Missing results on ImageNet**
> >
> > There were supposed to be numbers in the parenthesis in L435 (now L450). They should be there now. Thank you for catching this.

---

> > > ### Comment · Reviewer_a9mq · 2024-11-26
> > > **Thanks for the answers**
> > >
> > > Dear authors, thanks for addressing all concerns I brought up in my review. My rating for this paper was already positive, so after incorporating the feedback from the review process, I recommend it for acceptance.

---

### Official Review · Reviewer_7Zvh · 2024-11-04

**Soundness:** 3
**Presentation:** 2
**Contribution:** 2
**Rating:** 3
**Confidence:** 4

**Summary:**

In the research background, large-scale foundation models are common due to self-supervised learning techniques in deep learning, especially in computer vision. Cognitive psychology research indicates human visual perception is object-centric, leading to object-centric representation learning, though such models lack successful pre-training on large-scale real-world datasets. The research purpose is to propose the OCEBO method for pre-training object-centric models from scratch on real data to overcome limitations and unleash potential. The research methods involve a model architecture with an image encoder, slot attention encoder, slot decoder, and a target encoder of the same architecture, and a training objective formulated as a self-distillation bootstrapping problem with defined object-centric self-distillation loss including cross-view patch filtering and an optional mask sharpening stage. The experimental results on the MS COCO dataset and evaluation on multiple datasets with different metrics show that OCEBO can avoid slot collapse and achieve comparable performance to existing models with pre-trained target encoders while demonstrating good data scalability.

**Strengths:**

1. A new object-centric pre-training method, OCEBO, is proposed. It is the first self-distillation setup for training object-centric models from scratch on real-world data.

2. Experiments prove that OCEBO can avoid slot collapse and achieve performance comparable to existing methods using a large number of pre-trained images on multiple datasets while demonstrating good data scalability.

3. The importance of object-centric inductive biases is emphasized, and its positive impact on the target encoder is verified through experiments, providing new insights into the theory of object-centric learning.

**Weaknesses:**

1. Although good results have been achieved on the MS COCO dataset, the requirements for pre-training datasets are relatively high. Datasets containing simple scenes like ImageNet are not suitable for pre-training object-centric models, and a large-scale dataset suitable for pre-training object-centric models has not yet been found.

2. When comparing with existing state-of-the-art object-centric models, due to different pre-training methods and datasets used, the models are not directly comparable, which, to some extent, affects the accurate evaluation of model performance.

3. The experimental setup and evaluation system are still somewhat rudimentary and cannot fully demonstrate the scheme's advantages.

**Questions:**

1. When updating the target encoder as an exponential moving average (EMA) of the object-centric model encoder, how can we ensure that the introduced object-centric inductive biases do not overly affect the model's learning of other features, thus maintaining good generalization ability in different downstream tasks?
2. When the cross-view patch filtering method determines which patches to use for training, although it considers the feature quality of the target encoder, is it possible that this method may overlook some patch information that is potentially helpful for the model's learning? How can the accuracy and comprehensiveness of patch selection be better balanced?
3. The paper mentions that constructing a large-scale dataset suitable for pre-training object-centric models remains an open question. Do the authors have any preliminary ideas or directions on how to construct such a dataset?

---

> ### Author Response · Authors · 2024-11-20
> **Response to reviewer 7Zvh**
>
> Dear reviewer, thank you for a constructive review and in particular for appreciating the importance of explicit introduction of object-centric inductive biases and the novel insights into the theory of object-centric learning. Below we try to address your main concerns and we are happy to discuss anything in more detail.
>
> **Large-scale datasets (W1 and Q3)**
>
> In the paper, we stress that curated datasets with simple scenes such as ImageNet do not contain enough information for meaningful training of object-centric models. However, we believe that constructing a suitable dataset is not a large obstacle. In fact, we believe that any large-scale uncurated dataset might be enough to pretrain OCEBO. In the wild, scenes are rarely similar to those in ImageNet but are rather more complex with multiple (often interacting) objects, which is exactly what's necessary for successful pretraining of object-centric models.
>
> At the moment, we are already experimenting with uncurated large-scale datasets, such as LAION[1] and the Open Images Dataset[2]. As our computational resources are limited, we believe that releasing OCEBO into the community as is would be a valuable contribution as it would communicate to the community that pretraining from scratch is possible and will hopefully inspire others to join the efforts of scaling up object-centric models.
>
> **Slightly unfair comparison (W2 and W3)**
>
> We completely agree that due to significantly different training strategies and datasets, OCEBO is not directly comparable with other sota models. This is why we focus on emphasizing that OCEBO avoids slot collapse and is the first object-centric model pretrained from scratch rather than the exact numbers it achieves. As mentioned, our goal is to inspire the community to seek a paradigm shift by scaling up OCEBO and proposing novel pretraining strategies.
>
> Because of this, OCEBO uses the simplest design of object-centric learning components. For instance, replacing the MLP decoder with an autoregressive decoder from SPOT has been shown to have a significant impact on final performance. The same can be said for the high-resolution training stage introduced by FT-DINOSAUR. Incorporating those into OCEBO would surely increase its performance and bring the numbers closer to other sota methods, but we refrain from doing this as the evaluation, as you noted, would still be slightly unfair, and most importantly because this is not the main message of our work.
>
> **Ensuring good quality of patch-level representations (Q1 and Q2)**
>
> Great point! As the field of object-centric learning moves forward and we move towards unified global-patch-object level backbones, ensuring that all types of representations retain a good quality will become increasingly important. As in most other self-supervised methods, theoretically ensuring the quality of representations is difficult. However, in the case of OCEBO, we can experimentally verify that the quality of patch-level representations is not sacrificed at the expense of object-level representations.
>
> To this end, we evaluate OCEBO’s backbone in a dense task. We choose in-context semantic segmentation (or retrieval-based
> scene understanding) as described in Appendix A. We don’t directly compare to methods trained for in-context learning but we compare to CrOC, which is a patch-level representation learning approach. As you can note, the patch representations produced by OCEBO are on par with those of CrOC, indicating that we do not sacrifice the patch representation quality at the expense of object-centric representations.
>
> Additionally, we check the behavior of backbones of SPOT and FT-DINOSAUR before and after fine-tuning. It seems that both approaches sacrifice the backbone quality (which is quite expected given their reliance on non-object-centric backbones) but still keep it at reasonable levels.
>
> Finally, there is another dimension to your question. Although our object-centric objective can be viewed as a patch-level supervision signal, the features first go through a slot attention bottleneck and we filter out noisy patches, so the loss is less powerful than an explicit patch-level loss. Another way to ensure good patch-level representations more explicitly would be to introduce another self-distillation loss between patch representations of the object-centric model (before slot attention) and the teacher's features. We believe this would improve the numbers on dense downstream tasks at no expense for object-centric representations. If the time allows, we will aim to experimentally verify this before the end of the discussion period.
>
> [1] Schuhmann et al.: LAION-400M: Open Dataset of CLIP-Filtered 400 Million Image-Text Pairs. 2021
>
> [2] Kuznetsova et al.: The Open Images Dataset V4: Unified image classification, object detection, and visual relationship detection at scale. 2020.

---

### Author Response · Authors · 2024-11-20
**Revision summary**

Dear reviewers, thank you for very helpful and constructive reviews. The responses to your individual questions and concerns will follow shortly. Here, we summarize the changes made in the revised manuscript. The changes stem exclusively from the suggestions made by reviewers.

Reviewer 7Zvh
- Introduced Appendix A where OCEBO’s backbone is evaluated on a dense downstream task, ensuring that the patch-level representations are also of high quality.

Reviewer a9mq
- Added the quantitative measure of slot collapse to Section 4.2 and Table 1.
- Additional ablations of cross-view patch filtering added to Appendix B.
- Renamed loss in equation 1 (L212 and L226).
- Cited DINO in the paragraph prior to equations 5 and 6 (L240).
- Changed notation for nearest neighbors (L269).
- Introduced SPOT as the first to unfreeze the encoder (L072).
- Added the missing ImageNet results (L450).

Reviewer pa7a
- Added a more detailed scaling plot to Appendix C.

---

### Meta-Review · Area_Chair_BnQL · 2024-12-19

**Metareview:**

In this paper, the authors propose a self-distillation method for training object-centric models from scratch on real-world data. It uses an exponential moving average to update the target encoder. To prevent slot collapse, the authors introduce cross-view patch filtering, which selectively supervises informative patches. The approach demonstrates promising results on unsupervised object discovery benchmarks.

This paper was reviewed by four expert reviewers. After the rebuttal period, three of four reviewers are positive towards this paper. The only remaining concern comes from Reviewer 7Zvh, who also supported this paper initially. Their concern is about a closely related work [a], and I think this could be addressed by a minor revision. Therefore, the final decision is to accept this paper.

The authors are required to add a detailed discussion about [a] to their final version according to Reviewer 7Zvh's comments. The current version is not self-contained, as important details from paper [a] are not included. The authors briefly mention [a] in related works and preliminaries. However, these are not enough. The authors should highlight the differences between their method and [a], including the following aspects:
- Overall frameworks. The overall frameworks (Figure 1 in the submission and Figure 2 in [a]) are similar. Both frameworks leverage different views from a single image, as well as the Inverse Augmentation operations. Therefore, the technical differences should be carefully discussed.
- Loss designs, e.g., (1) Eq. 3 in the submission vs. Eq. 2 in [a]; and (2) Eq. 5 in the submission vs. Eq. 7 in [a].

The discussions with other reviewers should also be integrated into the final version as well.

[a] [*Self-Supervised Visual Representation Learning with Semantic Grouping*](https://arxiv.org/pdf/2205.15288), NeurIPS 2022.

**Additional Comments On Reviewer Discussion:**

This paper was reviewed by four expert reviewers. After the rebuttal period, three of four reviewers are positive towards this paper. The only remaining concern comes from Reviewer 7Zvh, who also supported this paper initially. Their concern is about a closely related work [a], and I think this could be addressed by a minor revision. Therefore, the final decision is to accept this paper.

---

### Decision · Program_Chairs · 2025-01-22

Accept (Poster)